# One-Pot Preparation of Mixed-Mode Reversed-Phase Anion-Exchange Silica Sorbent and its Application in the Detection of Cyclopiazonic Acid in Feeds and Agricultural Products

**DOI:** 10.3390/foods13101499

**Published:** 2024-05-12

**Authors:** Xuan Hu, Li Liu, Maomin Peng, Dan Zheng, Hong Xia, Youxiang Zhou, Lijun Peng, Xitian Peng

**Affiliations:** 1Institute of Agricultural Quality Standards and Testing Technology Research, Hubei Academy of Agricultural Sciences, Wuhan 430064, China; hxxuan0221@163.com (X.H.); liuli@hbaas.com (L.L.); pengmaomin@hbaas.com (M.P.); zhengdan@hbaas.com (D.Z.); xiahong@hbaas.com (H.X.); zhou_youxiang@aliyun.com (Y.Z.); 2Hubei Key Laboratory of Nutritional Quality and Safety of Agro-Products, Wuhan 430064, China

**Keywords:** cyclopiazonic acid, pyridine and octyl co-bonded silica, mixed-mode, agricultural products, UPLC-MS/MS

## Abstract

A novel co-bonded octyl and pyridine silica (OPS) sorbent was prepared and applied for the solid phase extraction (SPE) of cyclopiazonic acid (CPA, a type of mycotoxin) in feed and agricultural products for the first time. A simple mixed-ligand one-pot reaction strategy was employed for OPS sorbent preparation. Nitrogen adsorption–desorption measurements, elemental analysis (EI), thermal gravimetric analysis (TGA), and Fourier transform infrared spectroscopy (FT-IR) analysis demonstrated the successful immobilization of octyl and quaternary ammonium groups onto the surface of silica gel. The large specific surface area, high-density functional groups, and mixed-mode anion-exchange characteristics of these silica particles made them the ideal material for the efficient extraction of CPA. Additionally, the OPS sorbents displayed excellent batch-to-batch reproducibility, satisfactory reusability, and low cost. The SPE parameters were optimized to explore the ionic and hydrophobic interactions between CPA and the functional groups, and the ultra-high performance liquid chromatography coupled with triple-quadrupole tandem mass spectrometry (UPLC-MS/MS) parameters were optimized to obtain a desirable extraction efficiency and high sensitivity to CPA. Meanwhile, the OPS sorbent presented a satisfactory extraction selectivity and low matrix effect. Under the optimized conditions, our developed CPA detection method was used to determine CPA level in rice, wheat flour, corn flour, peanut, and feed samples, exhibiting a lower detection limit, better linearity, higher sensitivity, and satisfactory extraction recovery rate than previously reported methods. Therefore, our method can be preferentially used as a method for the detection of CPA in agricultural products and feeds.

## 1. Introduction

Mycotoxins, being secondary metabolites produced by fungal species, pose a great threat to human health due to their high carcinogenic, mutagenic, and immunosuppressive effects [1]. Statistical data from the Food and Agriculture Organization (FAO) indicates that approximately 25% of food crops worldwide are contaminated with mycotoxins before or after harvest, leading to significant economic losses and health concerns [2]. In general, cereals, oil crops, and their by-products such as feed are susceptible to contamination with mycotoxins. Some mycotoxins including aflatoxins (AFs) [3], fumonisins (FMs) [4], ochratoxins (OTs) [5], zearalenone (ZEN) [6], and their metabolites, have been widely studied due to their highly toxic effects on and serious pollution of many agricultural products. However, other mycotoxins such as cyclopiazonic acid (CPA) have not been thoroughly investigated.

CPA is a toxic metabolite produced by *Penicillium* and *Aspergillus* fungi. Under the suitable conditions, *Aspergillus flavus* strains can produce not only carcinogenic and teratogenic aflatoxins, but also large amounts of CPA [7]. CPA has been reported to cause damage to the liver, kidneys, pancreas, salivary glands, spleen, and cardiac and skeletal muscles of animals [8]. In the United States, a survey showed that 51% of corn and 90% of peanut samples were contaminated with CPA, with a maximum level of 2.9 mg/kg; in Europe, CPA concentration was found to be above 4 mg/kg in cheese due to the contamination by *Penicillium*; in India, CPA-contaminated millet led to human poisoning [9]. In addition, CPA is also found in various feeds and livestock and poultry meat products [10]. It has been reported that animals exhibit severe gastrointestinal upset and neurological disorders after ingesting feeds contaminated with CPA [8]. The CPA contamination of food and feed poses a great threat to human health.

Various methods such as capillary electrophoresis [11], enzyme-linked immunosorbent assay [7,12], high performance liquid chromatography [13,14,15,16], and liquid chromatography-tandem mass spectrometry (LC-MS/MS) [9,10,17,18,19,20] have been employed for the determination of CPA. Among these methods, LC-MS/MS stands out due to its high sensitivity, specificity, and simplified sample preparation requirements [10]. However, efficient sample preparation is necessary for the isolation of target compounds from the complex matrix prior to the instrumental analysis.

Solid phase extraction (SPE) is a well-developed clean-up technique with high pre-enrichment capability, low solvent consumption, easy automation, and availability of a range of distinct sorbents [21]. Generally, sorbents determine the selectivity and extraction efficiency of SPE. CPA is an indole tetramic and lipophilic monobasic acid with a pKa of 2.97 and a log *p* value of 3.83 [22], and it can interact with a mixed-mode sorbent through reversed-phase (RP)/ion-exchange (IE) interactions. The combination of RP and IE interaction mechanisms enables the selective extraction of target species and the elimination of matrix interference by altering the charge state of analytes and sorbents [23,24]. In recent years, some commercial silica- or polymer-based mixed-mode sorbents, such as Strata-X-A [25], Oasis WCX [26,27], and Bond Elut Certify II [28], have been used for the extraction of ionic compounds, exhibiting great performance. However, their specific surface functional groups and ion exchange capacity have yet to be modified or improved so as to enhance the performance of sorbents [23]. Accordingly, some non-commercial mixed-mode sorbents have been developed to improve the selectivity or capacity toward the target analytes. However, the synthetic methods of most reported adsorbents tend to involve multiple tedious steps [29,30,31].

Given this consideration, the objective of this study was to establish an effective SPE method for extracting CPA from food and feed samples, followed by detection using ultra-high performance liquid chromatography coupled with triple-quadrupole tandem mass spectrometry (UPLC-MS/MS). A simple one-pot preparation strategy was proposed to obtain mixed-mode reversed-phase anion-exchange (RP/AE) silica sorbent, and this RP/AE combination strategy can efficiently and selectively capture CPA from complex sample matrices. Meanwhile, a 13C-labelled internal standard was used to guarantee an accurate quantification. Finally, the proposed method was applied to the determination of CPA in rice, wheat flour, corn flour, and peanut, as well as their feed samples from the food market in Wuhan, China.

## 2. Experimental

### 2.1. Reagents and Materials

The 3-Chloropropyltrimethoxysilane and trimethoxyoctylsilane were purchased from Shanghai Aladdin Biochemical Technology Co., Ltd. (Shanghai, China). HPLC-grade acetone, methanol, and acetonitrile were obtained from J.T. Baker (Avantor Performance Materials, Mexico, Mexico). Silica gel (200–300 mesh) was supplied from Qingdao Ocean Chemical Factory (Qingdao, China). Anhydrous magnesium sulfate, sodium chloride, pyridine, and toluene were obtained from Shanghai Sinopharm Chemical Reagent Co., Ltd. (Shanghai, China). CPA and its isotopic internal standard [^13^C_20_]-CPA were obtained from Anpel laboratory Technologies Co., Ltd. (Shanghai, China) and Pribolab Bioengineering Co., Ltd. (Qingdao, China), respectively. The pure water used in this study was produced by a Milli-Q ultrapure water system (Milford, MA, USA).

### 2.2. Instrumentation

UPLC-MS/MS analysis was conducted using a Shimadzu LC-30A UPLC system (Tokyo, Japan) coupled to an AB Sciex 4500 triple quadrupole mass spectrometer (AB Sciex, Foster City, CA, USA) equipped with an IonDrive TurboV electrospray ionization (ESI) source. Fourier transform infrared spectroscopy (FT-IR) analysis was performed on a Thermo Scientific Nicolet 6700 (Emeryville, CA, USA). Thermal gravimetric analysis (TGA) was conducted on an STA-2500 thermal gravity analyzer (Netzsch, Germany). Elemental analysis was implemented on the vario EL cube elemental analyzer (Langenselbold, Germany). The nitrogen adsorption measurement (NAM) was carried out at 77 K on a Micromeritics ASAP 2460 analyzer (Norcross, GA, USA). The specific surface area of sorbents was calculated by the BET (Brunauer–Emmett–Teller) equation with P/P0 ranging between 0.05 and 0.3. The pore diameter and pore volume were calculated from the desorption branch of isotherm based on the BJH (Barrett–Joyner–Halenda) model.

### 2.3. Analytical Conditions

The CPA was separated with an UPLC@HSS T3 column (100 × 2.1 mm, 1.7 μm) (Waters, Zellik, Belgium). A binary mobile phase consisting of H_2_O with 0.01% formic acid (*v*/*v*) and 0.05% ammonium hydroxide (*v*/*v*) (phase A) and methanol (phase B) was employed as follows: 10% A for 0 min, 90% A for 6 min, 90% A for 9 min, 10% A for 9 min, and 10% A for 12 min. The column temperature was maintained at 40 °C.

The multiple reaction monitoring (MRM) analysis was performed in positive electrospray ionization (ESI) mode. Specifically, a CPA standard solution (0.2 μg/mL) was directly injected into the MS system to optimize the MS parameters, including precursor ion, product ion, declustering potential (DP), and collision energy (CE). Data were acquired with Analyst software (version 1.6.3) and processed with MultiQuant software (version 3.0.2). The MS working parameters were configured as follows: the ionspray voltage was set to −4500 V; the vaporizer temperature was maintained at 550 °C; the curtain gas, nebulizer gas, and auxiliary gas for the MS were all high-purity nitrogen with pressures set to 40, 55, and 55 psi, respectively.

### 2.4. Sample Collection

Peanut, rice, corn flour, and wheat flour samples were purchased from a food market in Wuhan, China. Feed samples were obtained from agricultural input stores in Wuhan, China. Feed, peanuts, and rice bulk samples were separately milled on an experimental miller, and then sieved with a 65-mesh sieve. Corn and wheat flour samples were directly used for analysis without any pre-processing. All samples were stored at 4 °C prior to analysis.

### 2.5. Preparation of Co-Bonded Octyl and Pyridine Silica (OPS) Sorbent

Herein, the mixed-mode OPS sorbent was prepared using a simple one-step procedure (Figure 1a). Specifically, silica was first activated by reflux and rehydroxylation in a hydrochloric acid–H_2_O (1:1) solution for 12 h, washed with H_2_O to reach a neutral status and then with acetone, and finally dried at 160 °C for 8 h. The 25.0 g of activated silica gel was suspended in 100 mL of toluene, into which 7.5 g of trimethoxychlorosilane, 13.3 g of trimethoxyoctylsilane, and 7.1 g of pyridine were added. The reaction was conducted at 110 °C for 20 h under nitrogen atmosphere to obtain the OPS sorbent. The resultant OPS sorbent was washed sequentially with toluene, MeOH, and acetone, and dried under vacuum at 60 °C overnight. 50.0 mg of the OPS sorbent was loaded into a 3 mL syringe and sandwiched between two thin polypropylene blocks. 

### 2.6. Sample Preparation

First, 1.0 g of sample was put into a 15 mL centrifuge tube. Then, 2 mL of water and 2 mL of acetonitrile with 0.5% acetic acid (*v*/*v*) were added. The mixture was vortexed for 1 min for CPA extraction. Subsequently, 0.4 g of sodium chloride and 1.6 g of anhydrous magnesium sulfate were added, and the mixture was vortexed again for 1 min. After centrifugation at 4500 rpm for 5 min, 1 mL of the CPA supernatant was mixed with 1 mL of water to obtain CPA sample solution for SPE cleanup.

As shown in Figure 1b, the OPS sorbents were first washed with 3 mL of acetone and 3 mL of 50% acetonitrile. Then, the CPA sample solution was passed through the SPE column containing OPS, and the SPE column was washed with 3 mL of 70% MeOH to remove the impurities absorbed by the OPS sorbents. Finally, the CPA absorbed by the OPS sorbents was eluted from the SPE column by 2 mL of methanol containing 5% ammonium hydroxide. The eluate was concentrated to less than 1 mL at 40 ℃ under N2 atmosphere, and then 50% methanol was added to reach a final volume of 1 mL for the UPLC-MS/MS analysis.

CPA-free samples (Peanut, rice, corn flour, and wheat flour) were selected for the spiked recovery test. Specifically, the CPA-free samples were spiked with CPA standard stock solutions to reach the appropriate concentration levels (2, 10, 50 μg/kg) and then pre-treated as described above.

## 3. Results and Discussion

### 3.1. Characterization of OPS Sorbent

NAM, FT-IR, TGA, and elemental analyses were performed to characterize the physicochemical properties of the obtained OPS sorbent. As shown in Figure 2a, a type IV adsorption isotherm and H1 hysteresis loop were observed, indicating the mesoporous structure of the OPS sorbent. The pore size of the OPS sorbent ranged from 2.2 nm to 15.1 nm with a mean value of 7.7 nm (Figure 2b). The BET surface area and pore volume of OPS sorbent was 297 m^2^/g and 0.63 cm^3^/g, respectively. These results indicated that the OPS sorbent had a large specific surface area and pore volume, which was favorable for the adsorption of target analytes.

The FTIR spectra of the pure silica and OPS sorbent are shown in Appendix A. Compared with pure silica, the OPS sorbent exhibits some new peaks in the spectrogram. The absorption peaks at 1460 and 1489 cm^−1^ might be attributed to stretching vibrations of the C=C and C=N bonds on the aromatic ring, respectively; and the absorption peaks at around 2900 cm^−1^ might be attributed to the stretching vibration of the C–H bonds, indicating the successful co-bonding of pyridine and n-octyl groups onto the surface of silica.

Elemental analysis indicated that the contents of nitrogen and carbon on the OPS sorbent were 0.46% and 9.29%, respectively. The surface coverage of the bonded octyl and pyridine species on the OPS sorbent was calculated according to the following formula by a previously reported method [32].
$$Coverage(\mu mol/m^{2})=\frac{X\%\times 10^{6}}{(AM)n100(1-X\%\frac{MW}{(AM)n100})S}$$
where *X*% represents the percentage increase in carbon or nitrogen content in the bonded support (determined by elemental analysis), *AM* denotes the atomic mass of carbon or nitrogen, *MW* represents the molecular weight of the species bonded to the silica surface, n is the number of carbon or nitrogen atoms present in the bonded species, and *S* signifies the specific surface area of the silica support (m^2^/g). Based on the contents of nitrogen and carbon, the surface coverage of pyridine and n-octyl groups were 1.2 μmol m^−2^ and 2.6 μmol m^−2^, respectively, with an approximate molar ratio of 1/2 of the pyridine to n-octyl groups.

Meanwhile, TGA was conducted to examine the content of functional groups. As depicted in Appendix A, pure silica demonstrated a slight mass loss with increasing temperature, whereas the OPS sorbent exhibited a more substantial mass loss, possibly attributed to the loss of the octyl and pyridine groups. TGA data at 800 °C indicated that the functional group content of OPS reached 8.9%.

### 3.2. Optimization of UPLC-MS/MS Parameters

The optimization of MS parameters was conducted using a pure CPA standard at a concentration of 0.2 μg/mL, which was directly delivered into the MS system using a built-in needle pump at a flow rate of 7 μL/min. According to the standards in the 2002/657/EC: Commission Decision [33], the multiple reaction monitoring (MRM) mode with at least two transitions using one precursor ion and two product ions was employed, in which the most intense MRM transition was selected for the quantitative MRM and the second most intense MRM transition was used for qualitative MRM. ACN/H_2_O (1/1, *v*/*v*) containing 5 mM NH4Ac was used as a typical mobile phase to identify protonated CPA molecules through the ESI. Under these conditions (the above mobile phase, 7 μL/min flow rate, and 0.2 μg/mL CPA concentration), the protonated CPA molecule (*m*/*z* 337, precursor ion) was isolated, and subjected to following collision-induced dissociation. Declustering potential (DP) ranging from 10 to 200 V and collision energy (CE) ranging from 5 to 100 V were employed to acquire product ions with the optimal response (Table 1). The fragment ions at *m*/*z* 196 and *m*/*z* 182 may be produced from the collision-induced dissociation (CID) process, which is indicated by the cleavage of the chemical bonds shown in Figure 3. The fragmentation pattern of CPA was consistent with a previous report on that of CPA in white mold cheese based on LC-MS analysis [17].

The chromatographic conditions were optimized to obtain the best separation with good peak shape. In light of the ionogenic nature of CPA, the polarity of both the stationary phase and the mobile phase was considered in the development of a chromatographic method. Firstly, CPA was separated with different types of UPLC C_18_ columns, including the Acquity BEH column (100 mm × 2.1 mm, 1.7 μm, Waters, Milford, MA, USA) and Acquity UPLC@HSS T3 column (100 mm × 2.1 mm, 1.7 μm, Waters, Ireland), respectively. Based on peak shape, total retention time, and chromatographic baseline, the Acquity UPLC@HSS T3 was selected as the suitable column for CPA detection.

Considering the low pKa value of CPA (about 2.97), different concentrations of formic acid (0.01% and 0.1%, *v*/*v*) was added to the mobile phase to improve the ionization efficiency of CPA. However, the observed chromatographic peak shape was very poor after formic acid addition (Figure 4a,b). This may be related to secondary ionic interactions between CPA and free silanol groups on the stationary phase surface [9]. Therefore, a certain amount of ammonium hydroxide was added as modifier of mobile phase to decrease the secondary interactions. As shown in Figure 4c, the peak shape and CPA detection sensitivity are significantly improved under the mobile phase of methanol and water mixed with ammonium hydroxide (0.05%, *v*/*v*) and formic acid (0.01%, *v*/*v*).

### 3.3. Optimization of SPE Parameters

To obtain a high extraction efficiency, several parameters potentially affecting CPA extraction performance were optimized, including the amount of OPS absorbent, pH, and NaCl concentration of the loading solution, solvent types and the amount used for washing the OPS column and eluting the CPA from the OPS column. A blank rice sample was spiked with 50 μg/kg CPA to optimize the SPE parameters.

#### 3.3.1. Optimization of Sorbent Amount

The mass of the sorbent is a key parameter influencing CPA extraction efficiency in SPE. The amount of OPS evaluated as a CPA extraction sorbent ranged from 30 to 100 mg. As shown in Figure 5a, the CPA recovery rate shows an increase when OPS sorbent is increased from 30 mg to 50 mg, but a gradual decrease when OPS is continuously increased from 50 to 100 mg. When a small amount of OPS absorbent was used, the extraction of CPA was incomplete, resulting in low recovery rate. However, excessive amounts of OPS sorbent would induce irreversible absorption, and thus more desorption solvents would be consumed to achieve effective desorption. Therefore, 50 mg was determined as the optimal sorbent amount for CPA extraction.

#### 3.3.2. Optimization of Loading Solution

The pH and NaCl concentration of the loading solution had an influence on the extraction of CPA, due to the ionic interaction between the quaternary ammonium groups of OPS and carboxyl groups of CPA. Consequently, NaCl was added to the sample solution within a mass concentration range of 0–30% to examine the impact of ionic concentration on the extraction of CPA. As shown in Figure 5b, the CPA extraction performance decreased sharply with the increasing NaCl concentrations, which might be attributed to the decrease in ionic interactions between the pyridine quaternary ammonium groups of OPS and carboxyl groups of CPA as the ion strength increased. Thus, no inorganic salt was added in the subsequent experiments to improve the extraction efficiency of CPA.

The pH of the sample solution affected the surface charge of the sorbent and the existing form of the CPA and thus affected the ion interaction between the sorbent and CPA. The effect of pH on the extraction performance of the OPS adsorbent was also investigated. As shown in Figure 5c, the extraction efficiency of CPA was slightly increased as the pH increased from 2.0 to 6.0, then slightly decreased to pH 7.0. With a CPA pKa value of 2.97, when the pH was increased from 2.0 to 6.0, CPA was gradually deprotonated, thus leading to the increase in ionic interaction. At pH 7.0, the residual silanol groups on the surface of the silica gel might be ionized thus decreasing the density of the positive charge on the surface of silica gel, eventually weakening ion exchange interaction between OPS adsorbent and CPA. Therefore, pH 6.0 of the sample solution was determined as the optimal pH value.

#### 3.3.3. Optimization of Washing Solution

Subsequently, washing solutions with different ratios of methanol and water were applied to eliminate the matrix interference while retaining the high extraction efficiency of CPA. As shown in Figure 5d, the recovery rate of CPA slowly decreased and remained above 90% when the concentration of methanol in washing solution increased from 0% to 70%. However, when the concentration of methanol continued to increase, the recovery rate of CPA dropped sharply. The possible reason might be that the hydrophobic and ionic interactions between OPS sorbent and CPA could all decrease with the increasing methanol concentration in washing solution and thus result in the decrease in CPA extraction efficiency. Therefore, 70% methanol was determined as the optimal washing solution.

#### 3.3.4. Optimization of Elution Solution

As mentioned above, there were mixed-mode reversed-phase anion-exchange interactions between the OPS sorbent and CPA. Thus, methanol containing 0% to 7% ammonia was utilized to disrupt the electrostatic and hydrophobic interactions between the OPS sorbent and CPA. As shown in Figure 5e, the extraction efficiency of CPA increased with the increasing ammonia concentration from 0% to 5%, and decreased when the ammonia concentration was 7%. This might be due to the instability of CPA at high ammonia concentration. Thus, methanol containing 5% ammonia was selected as the elution solution.

### 3.4. Reusability, Reproducibility, and Extraction Selectivity of OPS Sorbent

To investigate the reusability of OPS sorbent, several repetitive extraction/desorption cycles were carried out. In this case, the used OPS sorbent was employed for the subsequent SPE experiment under the optimal conditions. Additionally, three batches of OPS sorbents were independently prepared and used to extract CPA under the optimal SPE conditions. The results showed that the relative standard deviation (RSD) values were 9.8% and 8.5% for three cycles of repeated use of the same OPS sorbents and three different preparation batches of OPS sorbents, respectively, indicating the excellent reusability and reproducibility of the OPS sorbents for the CPA SPE process, which contributed to the reduction in cost to meet the needs for detection of CPA in real samples.

The extraction selectivity of OPS sorbent towards CPA was investigated by comparing extraction efficiency of CPA with that of other mycotoxins (including aflatoxin B1, deoxynivalenol, zearalenone, and fumonisin) and some pesticides (including 3-hydroxycarbofuran, acephate, acetamiprid, azoxystrobin, carbendazim, chlorbenzuron, dichlorvos, dimethoate, imidacloprid, paclobutrazol, phorate, propamocarb, and thiamethoxam) from a spiked rice sample under the same condition. The extraction efficiency of CPA was 84.2%, while that of other mycotoxins and pesticides was lower than 20%. In current study, the OPS adsorbent can have both reversed-phase and anion-exchange interactions with CPA, and mainly hydrophobic interactions with other toxins or pesticides, which greatly improves the selectivity towards CPA.

### 3.5. Method Validation

#### 3.5.1. Matrix Effect

In LC-MS/MS analysis, the ion intensity of target analytes could enhance or diminish because the co-extracted matrices might influence the ionization efficiency of the target analytes under ESI mode, which is the so-called matrix effect [34]. The matrix effect may affect the accuracy and repeatability of the analytic method. In this study, the matrix effects (*ME*) were evaluated according to the following equation:(1)$$ME(\%)=\frac{A_{matrix}-A_{solvent}}{A_{solvent}}\;\times \;100\%$$
in which A_matrix_ is the peak area of CPA in matrix, and A_solvent_ is the peak area of CPA in solvent. Generally, ME(%) from −20% to 20% can be considered as insignificant because such variability is close to the repeatability RSD values. ME(%) < 0 refers to ion inhibition effect, and ME(%) > 0 refers to ion enhancement effect. When −20 < ME(%) < 20, there was no matrix effect, and matrix matching calibration was not required for conventional analysis. Our data showed that the matrix effect of CPA in different agricultural products and feeds ranged from −3.0% to −12.9%, indicating a slight ion inhibition effect. It should be highlighted that the OPS sorbent is silica-based, which tends to perform fewer non-specific interactions with compounds than polymer-based Oasis MAX or Strata-X-A sorbents. In line with this, the OPS sorbent presented a low matrix effect and proved to be more selective than Oasis MAX or Strata-X-A sorbents. These results indicated that our proposed CPA SPE by OPS sorbent could effectively remove most of the interfering substances. 

#### 3.5.2. Linearity and Sensitivity

In this study, an isotopically labeled internal standard of CPA, [^13^C_20_]-CPA, was employed to improve the detection precision and decrease the matrix effects. The calibration curves of seven CPA concentration levels from 1 to 200 μg/kg were plotted by the ratio of the peak area of CPA/[^13^C_20_]-CPA. As shown in Table 2, the linearity range of CPA concentration was 1–200 μg/kg, and the correlation coefficient of the corresponding calibration curve was 0.9982. The detection of limit (LOD) and quantification of limit (LOQ) were calculated according to the CPA concentrations corresponding to the 3-fold and 10-fold signal-to-noise (S/N), respectively, which was 0.19 and 0.58 μg/kg, respectively.

#### 3.5.3. Specificity of CPA Detection

The specificity of CPA detection was examined by analyzing blank samples spiked with the target analyte. No interfering peaks were observed at or near the retention time of CPA. Following the regulations on pesticide residue analysis (SANTE/11312/2021), CPA was identified by comparing the retention time and MS/MS spectra in real samples with those in the standard samples. As illustrated in Table 3, the MRM transition ratios between the quantifier and qualifier ions in the various food samples spiked at 10 μg/kg and a standard solution were all within the tolerance range, indicating the high specificity of our proposed method. 

#### 3.5.4. Accuracy and Reproducibility

To evaluate the accuracy of the method, we performed recovery experiments using blank samples spiked with 2, 10, and 50 μg/kg CPA. The recovery rate was calculated as the ratio of detected CPA concentration using a calibration curve to the corresponding spiking concentration. Meanwhile, the intra-day precision was evaluated by comparing the six spiked parallel samples within one day, and inter-day precision was assessed by comparing the spiked samples prepared independently in six consecutive days. As it can be seen in Table 4, the recovery rate of CPA from different samples were within the range of 88.9–112.6%, and the intra-day and inter-day precisions were less than 8.2% and 10.9%, respectively, indicating the excellent accuracy and reproducibility of the proposed method.

### 3.6. Application of OPS Sorbent-Based SPE Method to the Detection of CPA in Real Agri-Products

Our proposed method was applied to real samples such as rice, flour, peanut, maize, and feed samples from the food market. To ensure the accuracy of the results, quality control of the samples was performed at intervals. The results showed that CPA was detected in four feed samples with a content of 1.7–37.8 μg/kg, and in a corn sample with a content of 3.8 μg/kg. These results indicated that our method is feasible, and it can be applied to the detection of CPA in agri-products. 

### 3.7. Method Comparison

To reveal the advantages of the proposed method, we compared this method with several reported methods for the detection of CPA in agri-products and feed samples in terms of sample types, pre-treatment method, determination technique, LOQs, recoveries, and RSDs. As shown in Table 5, the LOQ of our proposed method was roughly comparable to or better than other methods [3,8,29], but it was slightly higher than that of IMSPE/UPLC-MS [20]. The differences in CPA recovery between our method and the previously reported methods were minimal, and the RSDs of our method fell within the acceptable range. Meanwhile, compared with the commercially available mixed-mode sorbent such as Strata-X-A and Waters Oasis MAX, the one-pot preparation process of the OPS sorbent is very convenient. The cost of our homemade sorbent was relatively low since the raw materials for OPS sorbent preparation were much cheaper (less than one US dollar) than commercial SPE materials (more than three US dollars), thus allowing the OPS sorbent to be easily produced on large scale for broad use. More importantly, OPS demonstrated very good selectivity and extraction efficiency for CPA, which is promising for practical sample analysis. More importantly, it took less than 30 min for sample preparation by our proposed method, which was significantly more efficient than other reported methods.

## 4. Conclusions

In summary, a mixed-mode OPS sorbent was used to extract and purify CPA from agri-products and feeds. Specifically, a simple one-pot reaction strategy was employed to generate a mixed-mode reversed-phase anion-exchange silica sorbent. The combination of the anion exchange and reversed-phase interaction endowed the OPS sorbents with high specificity, selectivity, and efficiency for CPA extraction. Under optimized conditions, the proposed method exhibited good linearity, desirable sensitivity, excellent accuracy, and reproducibility. Additionally, the OPS sorbents displayed excellent batch-to-batch reproducibility and satisfactory reusability. Finally, our method was successfully applied to the detection of CPA in rice, flour, peanut, maize, and feed samples, indicating great application potential in real samples.

## Figures and Tables

**Figure 1 foods-13-01499-f001:**
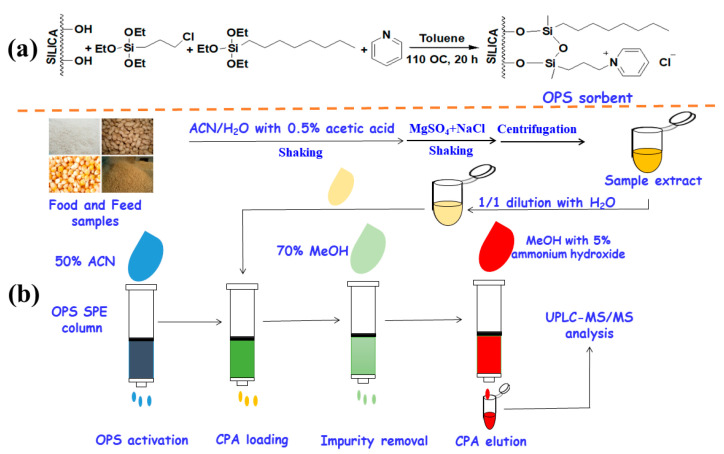
Flowchart of preparation of mixed-mode OPS sorbent (**a**) and its application in the extraction of CPA from feed and agricultural products (**b**).

**Figure 2 foods-13-01499-f002:**
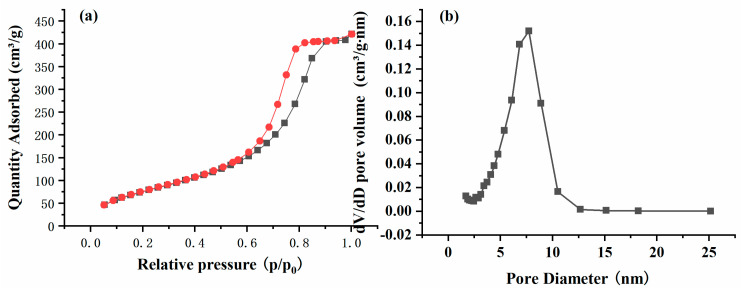
Nitrogen adsorption–desorption isotherm (**a**) and the pore size distribution curve (**b**) of OPS sorbent.

**Figure 3 foods-13-01499-f003:**
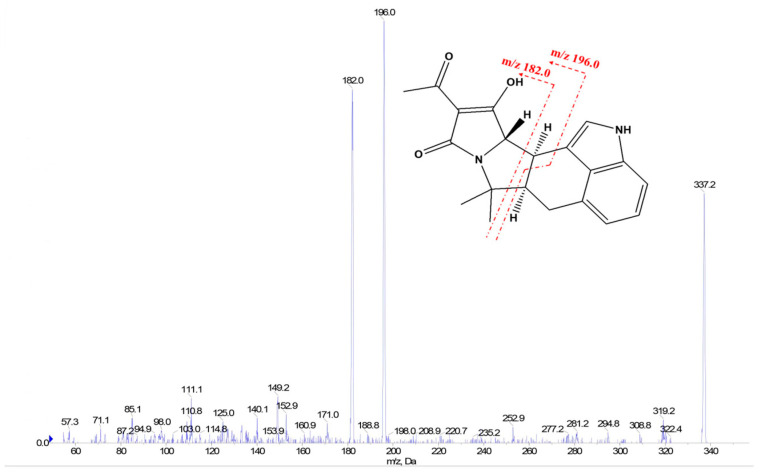
Product ion scan spectrum of CPA standard solution (200 ng/mL) and proposed structural origin of two MRM transitions.

**Figure 4 foods-13-01499-f004:**
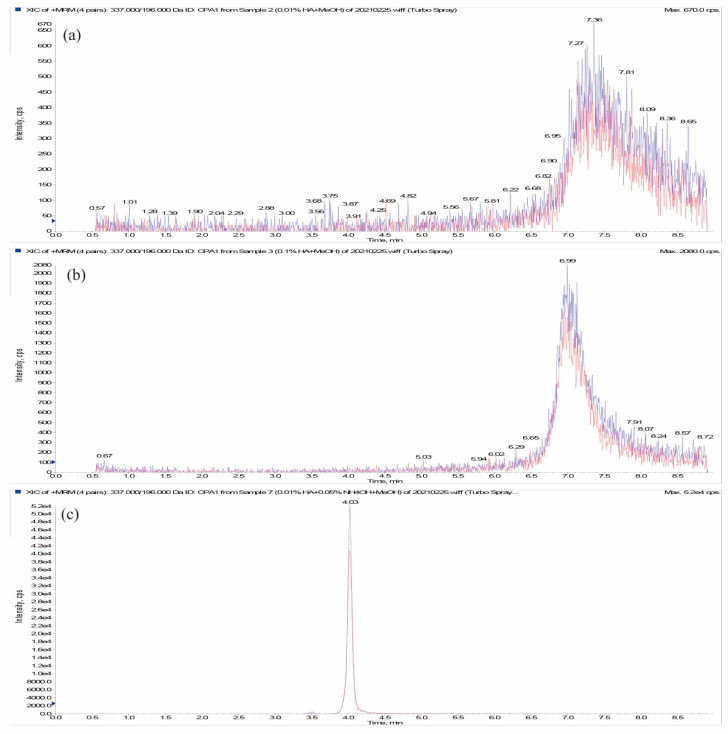
UPLC-MS/MS chromatograms of CPA standard (20 µg/L) under three mobile phases of methanol/water containing 0.01% formic acid (**a**), methanol/water containing 0.05% formic acid (**b**), methanol/water containing 0.01% formic acid and 0.05% ammonium hydroxide (**c**). Note: XeY means X × 10^Y^.

**Figure 5 foods-13-01499-f005:**
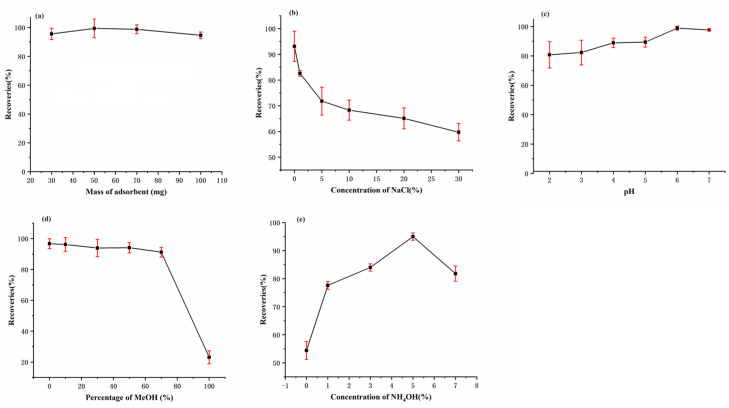
Effect of OPS sorbent amount (**a**), NaCl concentration (**b**), and pH value (**c**) of sample solution, MeOH concentration in washing solution (**d**), and NH4OH concentration in eluting solution (**e**) on CPA extraction efficiencies.

**Table 1 foods-13-01499-t001:** Retention time, quantification and qualification ion pairs, collision energies of CPA, and [^13^C_20_]-CPA by LC-MS/MS.

Analyte	Retention Time (min)	Quantification Ion (*m*/*z*)	CE1 (V)	DP1 (V)	Confirmation Ion (*m*/*z*)	CE2 (V)	DP2 (V)
CPA [^13^C_20_]-CPA	4.03	337 > 196	31	100	337 > 182	25	100
	4.02	357 > 210	31	100	/	/	/

**Table 2 foods-13-01499-t002:** Linear regression data of SPE of CPA.

Linear Range (μg/kg)	Regression Line				
	Slop	Intercept	R^2^ Value	LOD (μg/kg)	LOQ (μg/kg)
1–200	0.02521	0.00148	0.9982	0.19	0.58

Notes: LOD, detection of limit; and LOQ, quantification of limit.

**Table 3 foods-13-01499-t003:** Ratio between the quantifier and qualifier ions in standard solutions and spiked samples.

Samples	CPA			
	Ratio ^a^	Ratio ^b^	Deviation	Tolerance Range
Rice	76.4%	69.1%	9.5%	±30%
Wheat flour		73.2%	4.2%	
Corn flour		81.9%	7.2%	
Peanut		67.8%	11.3%	
Feed		82.3%	7.7%	

Notes: ^a^ Ratio between the quantifier and qualifier ions in standard solutions. ^b^ Ratio between the quantifier and qualifier ions in spiked oil samples.

**Table 4 foods-13-01499-t004:** Recovery rate and intra- and inter-day precisions for CPA detection in feed and food samples.

Matrix	Spiked Level (μg/kg)	Recovery Rate (%)	RSD_r_ (%, n = 6)	RSD_R_ (n = 6)
Rice	2	105.8	7.4	5.6
	10	112.4	5.8	7.0
	50	98.6	3.0	10.9
Wheat flour	2	89.5	4.6	8.5
	10	92.3	3.7	6.8
	50	105.4	8.2	5.9
Corn flour	2	112.6	2.0	8.8
	10	100.0	1.1	3.9
	50	88.9	4.6	8.3
Peanut	2	107.4	6.8	5.0
	10	101.9	1.3	4.4
	50	91.0	4.4	4.7
Feed	2	109.5	6.9	8.2
	10	88.9	5.4	9.3
	50	92.8	7.8	2.5

Notes: RSD_r_, relative standard deviation for intra-day precision; RSD_R_, inter-day precision.

**Table 5 foods-13-01499-t005:** Comparison of performances between our method and previously reported methods for CPA detection in food and feed samples.

Sample	Pretreatment Methods	Detection Method	Sorbent	Time for Sample Preparation	LOQ (μg/kg)	Recoveries (%)	RSDs (%)	References
Dry-fermented meat products	Improved QuECHERS combined with EMP lipid column cleanup	UPLC-MS/MS	EMP lipid	About 60 min	7.15	95.5–102.1	/	[10]
Wheat, peanut, rice, and Feed	Extraction with an alkaline methanol–water solution, defatting with hexane	HPLC-MS/MS	/	About 90 min	20	69–116	2–10	[9]
Maize and peanut	LLE	HPLC-UV	/	About 90 min	25	83.9–98.5	3.1–4.8	[35]
Wheat, peanut, and maize	IMSPE	UPLC-MS/MS	Immunosorbent	About 50 min	0.06	89.3–94.8	3.0–14.2	[20]
Rice, flour, peanut, maize, and feed	SPE	UPLC-MS/MS	Home-made OPS sorbent	<30 min	0.58	88.9–112.6	10.9	This study

## Data Availability

The original contributions presented in the study are included in the article, further inquiries can be directed to the corresponding author.

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
