# Peer review of "One-Pot Preparation of Mixed-Mode Reversed-Phase Anion-Exchange Silica Sorbent and its Application in the Detection of Cyclopiazonic Acid in Feeds and Agricultural Products"

_foods, 2024, doi:10.3390/foods13101499_

Round 1

Reviewer 1 Report

Comments and Suggestions for Authors

In the manuscript entitled " One-pot preparation of mixed-mode reversed-phase anion-exchange silica sorbent and its application to detection of cyclopiazonic acid in feeds and agricultural products a new sorbent called OPS, co-bonded octyl and pyridine silica was created and applied for the first time to extract cyclopiazonic acid (CPA), a type of mycotoxin, from feed and agricultural products. OPS sorbent was prepared using a simple mixed-ligand one-pot reaction method. Various analyses confirmed successful attachment of octyl and quaternary ammonium groups onto silica gel. OPS sorbent was characterized using NAM, FT-IR, and TGA techniques. UPLC-MS/MS parameters were optimized for better extraction efficiency and sensitivity. The developed method was used to detect CPA levels in rice, wheat flour,corn flour, peanut, and feed samples, showing improved performance compared to existing methods.

The study is comprehensive and well-structured. The text seems well-written, but here are a few suggestions for improvement:

line 66: “a log value of P 3.83” should have "a log P value" instead of "a log value of P" for clarity.

Line 143: It should be "overnight" instead of "for overnight" for correct phrasing.

Reviewer 2 Report

Comments and Suggestions for Authors

The manuscript describes an analytical method for the determination of cyclopiazonic acid (CPA) in feeds and agricultural products. It based on an extraction step that includes a tailor-made mixed-mode sorbent that interacts with the target analyte by both reversed-phase and anion-exchange mechanisms. Although the rationale of the work is interesting, the analytical approach described still lacks relevant information to conclude about its performance vs the current state of the art. Please find below an itemized list of comments that cover both general and detailed aspects of the manuscript.

1-   The context and comparison of the method with the current state of the art are insufficient. The authors have to compare the SPE method with similar sorbents commercially available (e.g. Oasis MAX from Waters). These sorbents are mentioned in the introduction but not used for comparison. The current comparison is insufficient to make conclusions about the pros and cons of the proposed method, especially because there are similar materials available in the market.

2-     The impurity removal description (line 157) is different from the information described in Fig. 1.

3-     The protocol used to spike the samples should be described in detail. This is not currently described.

4-     The authors should the experimental approach used for the method optimization. Why a Design of Experiments approach was not used?

5-     The authors should justify why is not desirable to extract other toxins along with CPA. Regarding that a separation method is used for the analyte detection, a multiclass sorbent may have some advantages.

6-     In the evaluation of the matrix effect, the authors mentioned an interval of ±20% as a tolerance criterion. This value has to be justified.

7-     The figures of merit should include limits of confidence whenever possible.

8-     What were the criteria used to select the samples used to perform the analysis? Why samples where CPA is frequent (e.g. cheese) were not selected?

9-     The abstract should include figures of merit and the main features of the proposed method.

Comments on the Quality of English Language

N/A

Reviewer 3 Report

Comments and Suggestions for Authors

This article reports total analytical study including the preparation of mixed-mode sorbent, and the application to the analysis of cyclopiazonic acid in agricultural products. The experimental results are well presented, and the reviewer considers that it should be published after minor revision.

1.     What is the advantage of the present OPS sorbent, over commercially available solid phase extraction media? This point of view is lacking, and the discussion on the issue should be added.

2.     Lines 192 to 914, “the surface coverage of pyridine and n-octyl groups were 1.2 micro mol m–2 and 2.6 micro mol m–2, respectively, with an approximate molar ratio 1/3 of pyridine to n-octyl groups”, are confusing. The molar ratio should be rather 1:2, not 1:3, if the value was calculated from 1.2 and 2.6 micro mol m–2.

3.     Figure 3 can be moved to supporting information.

4.     Chemical formulae, such as H2O, the number of atom(s) should be given in subscript style.
